# ADAPTIVE LEARNING OF TENSOR NETWORK STRUCTURES

## ABSTRACT

Tensor Networks (TN) offer a powerful framework to efficiently represent very high-dimensional objects. TN have recently shown their potential for machine learning applications and offer a unifying view of common tensor decomposition models such as Tucker, tensor train (TT) and tensor ring (TR). However, identifying the best tensor network structure from data for a given task is challenging. In this work, we leverage the TN formalism to develop a generic and efficient adaptive algorithm to jointly learn the structure and the parameters of a TN from data. Our method is based on a simple greedy approach starting from a rank one tensor and successively identifying the most promising tensor network edges for small rank increments. Our algorithm can adaptively identify TN structures with small number of parameters that effectively optimize any differentiable objective function. Experiments on tensor decomposition, tensor completion and model compression tasks demonstrate the effectiveness of the proposed algorithm. In particular, our method outperforms the state-of-the-art evolutionary topology search introduced in [19] for tensor decomposition of images (while being orders of magnitude faster) and finds efficient structures to compress neural networks outperforming popular TT based approaches [24].

## 1 INTRODUCTION

Matrix factorization is ubiquitous in machine learning and data science and forms the backbone of many algorithms. Tensor decomposition techniques emerged as a powerful generalization of matrix factorization. They are particularly suited to handle high-dimensional multi-modal data and have been successfully applied in neuroimaging [44], signal processing [3, 33], spatio-temporal analysis [1, 31] and computer vision [20]. Common tensor learning tasks include tensor decomposition (finding a low-rank approximation of a given tensor), tensor regression (which extends linear regression to the multi-linear setting), and tensor completion (inferring a tensor from a subset of observed entries).

Akin to matrix factorization, tensor methods rely on factorizing a high-order tensor into small factors. However, in contrast with matrices, there are many different ways of decomposing a tensor, each one giving rise to a different notion of rank, including CP, Tucker, Tensor Train (TT) and Tensor Ring (TR). For most tensor learning problems, there is no clear way of choosing which decomposition model to use, and the cost of model mis-specification can be high. It may even be the case that none of the commonly used models is suited for the task, and new decomposition models would achieve better tradeoffs between minimizing the number of parameters and minimizing a given loss function.

We propose an adaptive tensor learning algorithm which is agnostic to decomposition models. Our approach relies on the *tensor network* formalism, which has shown great success in the many-body physics community [28, 7, 6] and has recently demonstrated its potential in machine learning for compressing models [24, 39, 8, 23, 14, 41], developing new insights into the expressiveness of deep neural networks [4, 15], and designing novel approaches to supervised [35, 9] and unsupervised [34, 11, 22] learning. Tensor networks offer a unifying view of tensor decomposition models, allowing one to reason about tensor factorization in a general manner, without focusing on a particular model.

In this work, we design a greedy algorithm to efficiently search the space of tensor network structures for common tensor problems, including decomposition, completion and model compression. We start by considering the novel tensor optimization problem of minimizing a loss over arbitrary tensor network structures under a constraint on the number of parameters. To the best of our knowledge,

this is the first time that this problem is considered. The resulting problem is a bi-level optimization problem where the upper level is a discrete optimization over tensor network structures, and the lower level is a continuous optimization of a given loss function. We propose a greedy approach to optimize the upper-level problem, which is combined with continuous optimization techniques to optimize the lower-level problem. Starting from a rank one initialization, the greedy algorithm successively identifies the most promising edge of a tensor network for a rank increment, making it possible to adaptively identify from data the tensor network structure which is best suited for the task at hand.

The greedy algorithm we propose is conceptually simple, and experiments on tensor decomposition, completion and model compression tasks showcase its effectiveness. Our algorithm significantly outperforms a recent evolutionary algorithm [19] for tensor network decomposition on an image compression task by discovering structures that require less parameters while simultaneously achieving lower recovery errors. The greedy algorithm also outperforms CP, Tucker, TT and TR algorithms on an image completion task and finds more efficient TN structures to compress fully connected layers in neural networks than the TT based method introduced in [24].

**Related work**    Adaptive tensor learning algorithms have been previously proposed, but they only consider determining the rank(s) of a specific decomposition and are often tailored to a specific tensor learning task (e.g., decomposition or regression). In [1], a greedy algorithm is proposed to adaptively find the ranks of a Tucker decomposition for a spatio-temporal forecasting task, and in [38] an adaptive Tucker based algorithm is proposed for background subtraction. In [42], the authors present a Bayesian approach for automatically determining the rank of a CP decomposition. In [2] an adaptive algorithm for tensor decomposition in the hierarchical Tucker format is proposed. In [10] a stable rank-adaptive alternating least square algorithm is introduced for completion in the TT format. The problem we consider is considerably more general since we do not assume a fixed tensor network structure (e.g. Tucker, TT, CP, etc.). Exploring other decomposition relying on the tensor network formalism has been sporadically explored. The work which is the most closely related to our contribution is [19] where evolutionary algorithms are used to approximate the best tensor network structure to exactly decompose a given target tensor. However, the method proposed in [19] only searches for TN structures with uniform ranks (with the rank being a hyperparameter) and is limited to the problem of tensor decomposition. In contrast, our method is the first to jointly explore the space of structures and (non-uniform) ranks to minimize an arbitrary loss function over the space of tensor parameters. Lastly, [12] proposes to explore the space of tensor network structures for compressing neural networks, a rounding algorithm for general tensor networks is proposed in [21] and the notions of rank induced by arbitrary tensor networks are studied in [40].

## 2    PRELIMINARIES

In this section, we present notions of tensor algebra and tensor networks. We first introduce notations. For any integer $k$, $[k]$ denotes the set of integers from 1 to $k$. We use lower case bold letters for vectors (e.g. $\mathbf{v} \in \mathbb{R}^{d_1}$), upper case bold letters for matrices (e.g. $\mathbf{M} \in \mathbb{R}^{d_1 \times d_2}$) and bold calligraphic letters for higher order tensors (e.g. $\boldsymbol{\mathcal{T}} \in \mathbb{R}^{d_1 \times d_2 \times d_3}$). The $i$th row (resp. column) of a matrix $\mathbf{M}$ will be denoted by $\mathbf{M}_{i,:}$ (resp. $\mathbf{M}_{:,i}$). This notation is extended to slices of a tensor in the obvious way.

**Tensors and tensor networks**    We first recall basic definitions of tensor algebra; more details can be found in [17]. A *tensor* $\boldsymbol{\mathcal{T}} \in \mathbb{R}^{d_1 \times \cdots \times d_p}$ can simply be seen as a multidimensional array $(\boldsymbol{\mathcal{T}}_{i_1, \cdots, i_p} \ : \ i_n \in [d_n], n \in [p])$. The inner product of two tensors is defined by $\langle \boldsymbol{\mathcal{S}}, \boldsymbol{\mathcal{T}} \rangle = \sum_{i_1, \cdots, i_p} \boldsymbol{\mathcal{S}}_{i_1 \cdots i_p} \boldsymbol{\mathcal{T}}_{i_1 \cdots i_p}$ and the Frobenius norm of a tensor is defined by $\|\boldsymbol{\mathcal{T}}\|_F^2 = \langle \boldsymbol{\mathcal{T}}, \boldsymbol{\mathcal{T}} \rangle$. The *mode-$n$ matrix product* of a tensor $\boldsymbol{\mathcal{T}}$ and a matrix $\mathbf{X} \in \mathbb{R}^{m \times d_n}$ is a tensor denoted by $\boldsymbol{\mathcal{T}} \times_n \mathbf{X}$. It is of size $d_1 \times \cdots \times d_{n-1} \times m \times d_{n+1} \times \cdots \times d_p$ and is obtained by contracting the $n$th mode of $\boldsymbol{\mathcal{T}}$ with the second mode of $\mathbf{X}$, e.g. for a 3rd order tensor $\boldsymbol{\mathcal{T}}$, we have $(\boldsymbol{\mathcal{T}} \times_2 \mathbf{X})_{i_1 i_2 i_3} = \sum_j \boldsymbol{\mathcal{T}}_{i_1 j i_3} \mathbf{X}_{i_2 j}$. The $n$th mode matricization of $\boldsymbol{\mathcal{T}}$ is denoted by $\boldsymbol{\mathcal{T}}_{(n)} \in \mathbb{R}^{d_n \times \prod_{i \neq n} d_i}$. *Tensor network diagrams* allow one to represent complex operations on tensors in a graphical and intuitive way. A tensor network (TN) is simply a graph where nodes represent tensors, and edges represent contractions between tensor modes, i.e. a summation over an index shared by two tensors. In a tensor network, the arity of a vertex (i.e. the number of *legs* of a node) corresponds to the order of the tensor (see Figure 1). Connecting two legs in a tensor network represents a contraction over the corresponding indices. Consider

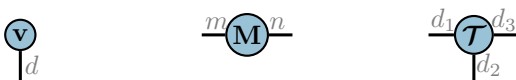

Figure 1: Tensor network representation of a vector $\mathbf{v} \in \mathbb{R}^d$, a matrix $\mathbf{M} \in \mathbb{R}^{m \times n}$ and a tensor $\mathcal{T} \in \mathbb{R}^{d_1 \times d_2 \times d_3}$.

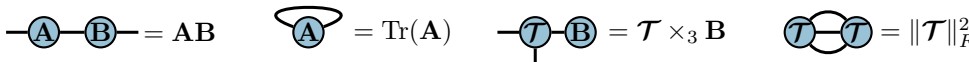

Figure 2: Tensor network representation of common operation on matrices and tensors.

the following simple tensor network with two nodes: $\overset{m}{-}\!\!\boxed{A}\!\!\overset{n}{-}\!\!\boxed{x}$. The first node represents a matrix $\mathbf{A} \in \mathbb{R}^{m \times n}$ and the second one a vector $\mathbf{x} \in \mathbb{R}^n$. Since this tensor network has one dangling leg (i.e. an edge which is not connected to any other node), it represents a first order tensor, i.e., a vector. The edge between the second leg of $\mathbf{A}$ and the leg of $\mathbf{x}$ corresponds to a contraction between the second mode of $\mathbf{A}$ and the first mode of $\mathbf{x}$. Hence, the resulting tensor network represents the classical matrix-vector product, which can be seen by calculating the $i$th component of this tensor network: $i\!-\!\boxed{A}\!-\!\boxed{x} \; = \sum_j \mathbf{A}_{ij}\mathbf{x}_j = (\mathbf{Ax})_i$ . Other examples of tensor network representations of common operations on matrices and tensors can be found in Figure 2. Lastly, it is worth mentioning that disconnected tensor networks correspond to tensor products, e.g., $-\!\boxed{u}$ $\boxed{v}\!- \; = \mathbf{uv}^\top$ is the outer product of $\mathbf{u}$ and $\mathbf{v}$ with components $i\!-\!\boxed{u}$ $\boxed{v}\!-\!j = \mathbf{u}_i\mathbf{v}_j$ . Consequently, an edge of dimension (or rank) 1 in a TN is equivalent to having no edge between the two nodes, e.g., if $R = 1$ we have $i\!-\!\boxed{A}\!\overset{R}{-}\!\boxed{B}\!-\!j \; = \sum_{r=1}^R \mathbf{A}_{i,r}\mathbf{B}_{r,j} = \mathbf{A}_{i,1}\mathbf{B}_{1,j} = \; i\!-\!\boxed{A} \; \boxed{B}\!-\!j$.

**Tensor decomposition and tensor rank** We now briefly present the most common tensor decomposition models, omitting the CP decomposition which cannot be described using the TN formalism unless hyper-edges are allowed (which we do not consider in this work). For the sake of simplicity we consider a fourth order tensor $\mathcal{T} \in \mathbb{R}^{d_1 \times d_2 \times d_3 \times d_4}$, each decomposition can be straightforwardly extended to higher-order tensors. A Tucker decomposition [36] decomposes $\mathcal{T}$ as the product of a core tensor $\mathcal{G} \in \mathbb{R}^{R_1 \times R_2 \times R_3 \times R_4}$ with four factor matrices $\mathbf{U}_i \in \mathbb{R}^{d_i \times R_i}$ for $i = 1, \cdots, 4$: $\mathcal{T} = \mathcal{G} \times_1 \mathbf{U}_1 \times_2 \mathbf{U}_2 \times_3 \mathbf{U}_3 \times_4 \mathbf{U}_4$. The Tucker rank, or multilinear rank, of $\mathcal{T}$ is the smallest tuple $(R_1, R_2, R_3, R_4)$ for which such a decomposition exists. The tensor ring (TR) decomposition [43, 25, 29] expresses each component of $\mathcal{T}$ as the trace of a product of slices of four core tensors $\mathcal{G}^{(1)} \in \mathbb{R}^{R_0 \times d_1 \times R_1}$, $\mathcal{G}^{(2)} \in \mathbb{R}^{R_1 \times d_2 \times R_2}$, $\mathcal{G}^{(3)} \in \mathbb{R}^{R_2 \times d_3 \times R_3}$ and $\mathcal{G}^{(4)} \in \mathbb{R}^{R_4 \times d_4 \times R_0}$: $\mathcal{T}_{i_1,i_2,i_3,i_4} = \mathrm{Tr}(\mathcal{G}^{(1)}_{:,i_1,:}\mathcal{G}^{(2)}_{:,i_2,:}\mathcal{G}^{(3)}_{:,i_3,:}\mathcal{G}^{(4)}_{:,i_4,:})$. The tensor train (TT) decomposition [26] (also known as matrix product states in the physics community) is a particular case of the tensor ring decomposition where $R_0$ must be equal to 1 ($R_0$ is thus omitted when referring to the rank of a TT decomposition). Similarly to Tucker, the TT and TR decompositions naturally give rise to an associated notion of rank: the TR rank (resp. TT rank) is the smallest tuple $(R_0, R_1, R_2, R_3)$ (resp. $(R_1, R_2, R_3)$) such that a TR (resp. TT) decomposition exists.

Tensor networks offer a unifying view of tensor decomposition models: Figure 3 shows the TN representation of common models. Each decomposition is naturally associated with the graph topology of the underlying TN. For example, the Tucker decomposition corresponds to star graphs, the TT decomposition corresponds to chain graphs, and the TR decomposition model corresponds to cyclic graphs. The relation between the rank of a decomposition and its number of parameters is different for each model. Letting $p$ be the order of the tensor, $d$ its largest dimension and $R$ the rank of the decomposition (assuming uniform ranks), the number of parameters is in $\mathcal{O}(R^p + pdR)$ for Tucker, and $\mathcal{O}(pdR^2)$ for TT and TR. One can see that the Tucker decomposition is not well suited for tensors of very high order since the size of the core tensor grows exponentially with $p$.

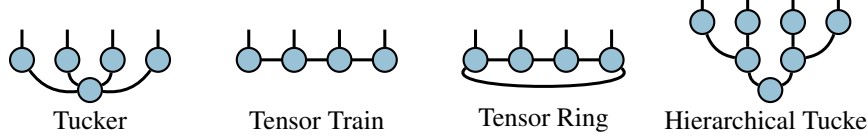

Figure 3: Tensor network representation of common decomposition models for a 4th order tensor.

## 3 A GREEDY ALGORITHM FOR TENSOR NETWORK STRUCTURE LEARNING

### 3.1 TENSOR NETWORK OPTIMIZATION

We consider the problem of minimizing a loss function $\mathcal{L} : \mathbb{R}^{d_1 \times \cdots \times d_p} \to \mathbb{R}_+$ w.r.t. a tensor $\mathcal{W}$ efficiently parameterized as a tensor network (TN). We first introduce our notations for TN.

Without loss of generality, we consider TN having one factor per dimension of the parameter tensor $\mathcal{W} \in \mathbb{R}^{d_1 \times \cdots \times d_p}$, where each of the factors has one dangling leg corresponding to one of the dimensions $d_i$ (we will discuss how this encompasses TN structures with internal nodes such as Tucker at the end of this section). In this case, a TN structure is summarized by a collection of ranks $(R_{i,j})_{1 \le i < j \le p}$ where each $R_{i,j} \ge 1$ is the dimension of the edge connecting the $i$th and $j$th nodes of the TN (for convenience, we assume $R_{i,j} = R_{j,i}$ if $i > j$). If there is no edge between nodes $i$ and $j$ in a TN, $R_{i,j}$ is thus equal to 1 (see Section 2). A TN decomposition of $\mathcal{W} \in \mathbb{R}^{d_1 \times \cdots \times d_p}$ is then given by a collection of core tensors $\mathcal{G}^{(1)}, \cdots, \mathcal{G}^{(p)}$ where each $\mathcal{G}^{(i)}$ is of size $R_{1,i} \times \cdots \times R_{i-1,i} \times d_i \times R_{i,i+1} \times \cdots \times R_{i,p}$. Each core tensor is of order $p$ but some of its dimensions may be equal to one (representing the absence of edge between the two cores in the TN structure). We use $\mathrm{TN}(\mathcal{G}^{(1)}, \cdots, \mathcal{G}^{(p)})$ to denote the resulting tensor. Formally, for an order 4 tensor we have

$$\mathrm{TN}(\mathcal{G}^{(1)}, \cdots, \mathcal{G}^{(4)})_{i_1 i_2 i_3 i_4} = \sum_{j_1^2=1}^{R_{1,2}} \sum_{j_1^3=1}^{R_{1,3}} \cdots \sum_{j_3^4=1}^{R_{3,4}} \mathcal{G}^{(1)}_{i_1, j_1^2, j_1^3, j_1^4} \mathcal{G}^{(2)}_{j_1^2, i_2, j_2^3, j_2^4} \mathcal{G}^{(3)}_{j_1^3, j_2^3, i_3, j_3^4} \mathcal{G}^{(4)}_{j_1^4, j_2^4, j_3^4, i_4}.$$

This definition is straightforwardly extended to TN representing tensors of arbitrary orders.

As an illustration, for a TT decomposition the ranks of the tensor network representation would be such that $R_{i,j} \neq 1$ if and only if $j = i + 1$. The problem of finding a rank $(r_1, r_2, r_3)$ TT decomposition of a target tensor $\mathcal{T} \in \mathbb{R}^{d_1 \times d_2 \times d_3 \times d_4}$ can thus be formalized as

$$\min_{\substack{\mathcal{G}^{(1)} \in \mathbb{R}^{d_1 \times r_1 \times 1 \times 1}, \mathcal{G}^{(2)} \in \mathbb{R}^{r_1 \times d_2 \times r_2 \times 1}, \\ \mathcal{G}^{(3)} \in \mathbb{R}^{1 \times r_2 \times d_3 \times r_3}, \mathcal{G}^{(4)} \in \mathbb{R}^{1 \times 1 \times r_3 \times d_4}}} \mathcal{L}(\mathrm{TN}(\mathcal{G}^{(1)}, \mathcal{G}^{(2)}, \mathcal{G}^{(3)}, \mathcal{G}^{(4)})) \tag{1}$$

where $\mathcal{L}(\mathcal{W}) = \|\mathcal{T} - \mathcal{W}\|_F^2$. Other common tensor problems can be formalized in this manner. For example, the tensor train completion problem would be formalized similarly with the loss function being $\mathcal{L}(\mathcal{W}) = \frac{1}{|\Omega|} \sum_{(i_1, \cdots, i_p) \in \Omega} (\mathcal{W}_{i_1, \cdots, i_p} - \mathcal{T}_{i_1, \cdots, i_p})^2$ where $\Omega \subset [d_1] \times \cdots \times [d_p]$ is the set of observed entries of $\mathcal{T} \in \mathbb{R}^{d_1 \times \cdots \times d_p}$, and learning TT models for classification [35] and sequence modeling [11] also falls within this general formulation by using the cross-entropy or log-likelihood as a loss function.

We now explain how our formalism encompasses TN structure with internal nodes, such as the Tucker format. Since a rank one edge in a TN is equivalent to having no edge, internal cores can be represented as cores whose dangling leg have dimension 1. Consider for example the Tucker decomposition $\mathcal{T} = \mathcal{G} \times_1 \mathbf{U}_1 \times_2 \mathbf{U}_2 \times_3 \mathbf{U}_3 \in \mathbb{R}^{d_1 \times d_2 \times d_3}$ of rank $(r_1, r_2, r_3)$. The tensor $\mathcal{T}$ can naturally be seen as a fourth order tensor $\tilde{\mathcal{T}} \in \mathbb{R}^{1 \times d_1 \times d_2 \times d_3}$, $\mathcal{G}$ as $\tilde{\mathcal{G}} \in \mathbb{R}^{1 \times r_1 \times r_2 \times r_3}$, $\mathbf{U}_1$ as $\tilde{\mathbf{U}}_1 \in \mathbb{R}^{r_1 \times d_1 \times 1 \times 1}$, $\mathbf{U}_2$ as $\tilde{\mathbf{U}}_2 \in \mathbb{R}^{r_2 \times 1 \times d_2 \times 1}$ and $\mathbf{U}_3$ as $\tilde{\mathbf{U}}_3 \in \mathbb{R}^{r_3 \times 1 \times 1 \times d_3}$. With these definitions, one can check that $\mathrm{TN}(\tilde{\mathcal{G}}, \tilde{\mathbf{U}}_1, \tilde{\mathbf{U}}_2, \tilde{\mathbf{U}}_3) = \mathcal{G} \times_1 \mathbf{U}_1 \times_2 \mathbf{U}_2 \times_3 \mathbf{U}_3 = \mathcal{T}$. More complex TN structure with internal nodes such as hierarchical Tucker can be represented using our formalism in a similar way. The assumption that each core tensor in a TN structure has one dangling leg corresponding to each of the dimensions of the tensor $\mathcal{T}$ is thus without loss of generality, since it suffices to augment $\mathcal{T}$ with singleton dimensions to represent TN structures with internal nodes.

## 3.2 PROBLEM STATEMENT

A large class of TN learning problems consist in optimizing a loss function w.r.t. the core tensors of a *fixed* TN structure; this is, for example, the case of the TT completion problem: the rank of the decomposition may be selected using, e.g., cross-validation, but the *overall structure* of the TN is fixed *a priori*. In contrast, we propose to optimize the loss function simultaneously w.r.t. the core tensors of the TN *and the TN structure itself*. This joint optimization problem can be formalized as

$$\min_{\substack{R_{i,j}, \\ 1 \leq i < j \leq p}} \min_{\boldsymbol{\mathcal{G}}^{(1)}, \cdots, \boldsymbol{\mathcal{G}}^{(p)}} \mathcal{L}(\text{TN}(\boldsymbol{\mathcal{G}}^{(1)}, \cdots, \boldsymbol{\mathcal{G}}^{(p)})) \quad \text{s.t. } \texttt{size}(\boldsymbol{\mathcal{G}}^{(1)}, \cdots, \boldsymbol{\mathcal{G}}^{(p)}) \leq C \qquad (2)$$

where $\mathcal{L}$ is a loss function, each core tensor $\boldsymbol{\mathcal{G}}^{(i)}$ is in $\mathbb{R}^{R_{1,i} \times \cdots \times R_{i-1,i} \times d_i \times R_{i,i+1} \times \cdots \times R_{i,p}}$, $C$ is a bound on the number of parameters, and $\texttt{size}(\boldsymbol{\mathcal{G}}^{(1)}, \cdots, \boldsymbol{\mathcal{G}}^{(p)})$ is the number of parameters of the TN, which is equal to $\sum_{i=1}^{p} d_i R_{1,i} \cdots R_{i-1,i} R_{i,i+1} \cdots R_{i,p}$. Note that if $K$ is the maximum arity of a node in a TN, its number of parameters is in $\mathcal{O}\left(pdR^K\right)$ where $d = \max_i d_i$ and $R = \max_{i,j} R_{i,j}$.

Problem 2 is a bi-level optimization problem where the upper level is a discrete optimization over TN structures, and the lower level is a continuous optimization problem (assuming the loss function is continuous). If it is possible to solve the lower level continuous optimization, an exact solution can be found by enumerating the search space of the upper level, i.e. enumerating all TN structures satisfying the constraint on the number of parameters, and selecting the one achieving the lower value of the objective. This approach is, of course, not realistic since the search space is combinatorial in nature, and its size will grow exponentially with $p$. Moreover, for most tensor learning problems, the lower-level continuous optimization problem is NP-hard [13]. In the next section, we propose a general greedy approach to tackle this problem.

## 3.3 GREEDY ALGORITHM

We propose a greedy algorithm which consists in first optimizing the loss function $\mathcal{L}$ starting from a rank one initialization of the tensor network, i.e. $R_{i,j}$ is set to one for all $i, j$ and each core tensor $\boldsymbol{\mathcal{G}}^{(i)} \in \mathbb{R}^{R_{1,i} \times \cdots \times R_{i-1,i} \times d_i \times R_{i,i+1} \times \cdots \times R_{i,p}}$ is initialized randomly. At each iteration of the greedy algorithm, the most promising edge of the current TN structure is identified through some efficient heuristic, the corresponding rank is increased, and the loss function is optimized w.r.t. the core tensors of the new TN structure initialized through a weight transfer mechanism. In addition, at each iteration, the greedy algorithm identifies nodes that can be split to create internal nodes in the TN structure by analyzing the spectrum of matricizations of its core tensors.

The overall greedy algorithm, named `Greedy-TN`, is summarized in Algorithm 1. In the remaining of this section, we describe the continuous optimization, weight transfer, best edge identification and node splitting procedures. For Problem 2, a natural stopping criterion for the greedy algorithm is when the maximum number of parameters is reached, but more sophisticated stopping criteria can be used. For example, the algorithm can be stopped once a given loss threshold is reached, which leads to an approximate solution to the problem of identifying the TN structure with the least number of parameters achieving a given loss threshold. For learning tasks (e.g., TN classifiers or tensor completion), the stopping criterion can be based on validation data (e.g., using early stopping).

**Continuous Optimization**   Assuming that the loss function $\mathcal{L}$ is continuous and differentiable, standard gradient-based optimization algorithms can be used to solve the inner optimization problem (line 12 of Algo. 1). E.g., in our experiments on compressing neural network layers (see Section 4) we use Adam [16]. For particular losses, more efficient optimization methods can be used: in our experiments on tensor completion and tensor decomposition, we use the Alternating Least-Squares (ALS) [17, 5] algorithm which consists in alternatively solving the minimization problem w.r.t. one of the core tensors while keeping the other ones fixed until convergence.

**Weight Transfer**   A key idea of our approach is to restart the continuous optimization process from the previous iteration of the greedy algorithm: we initialize the new slices of the two core tensors connected by the incremented edge to values close to 0, while keeping all the other parameters of the TN unchanged (line 8-10 of Algo. 1). E.g., for a tensor network of order 4, increasing the rank of the edge $(1, 2)$ by 1 is done by adding a slice of size $d_1 \times R_{1,3} \times R_{1,4}$ (resp. $d_2 \times R_{2,3} \times R_{2,4}$) to the second mode of $\boldsymbol{\mathcal{G}}^{(1)}$ (resp. first mode of $\boldsymbol{\mathcal{G}}^{(2)}$). After this operation, the new shape of $\boldsymbol{\mathcal{G}}^{(1)}$ will

---

**Algorithm 1** `Greedy-TN`: Greedy algorithm for tensor network structure learning.

---

**Input:** Loss function $\mathcal{L} : \mathbb{R}^{d_1 \times \cdots \times d_p} \to \mathbb{R}$, splitting node threshold $\varepsilon$.
1: // *Initialize tensor network to a random rank one tensor and optimize loss function.*
2: $R_{i,j} \leftarrow 1$ for $1 \le i < j \le p$
3: Initialize core tensors $\boldsymbol{\mathcal{G}}^{(i)} \in \mathbb{R}^{R_{1,i} \times \cdots \times R_{i-1,i} \times d_i \times R_{i,i+1} \times \cdots \times R_{i,p}}$ randomly
4: $(\boldsymbol{\mathcal{G}}^{(1)}, \cdots, \boldsymbol{\mathcal{G}}^{(p)}) \leftarrow$ `optimize` $\mathcal{L}(\text{TN}(\boldsymbol{\mathcal{G}}^{(1)}, \cdots, \boldsymbol{\mathcal{G}}^{(p)}))$ w.r.t. $\boldsymbol{\mathcal{G}}^{(1)}, \cdots, \boldsymbol{\mathcal{G}}^{(p)}$
5: **repeat**
6:    $(i, j) \leftarrow$ `find-best-edge`$(\mathcal{L}, (\boldsymbol{\mathcal{G}}^{(1)}, \cdots, \boldsymbol{\mathcal{G}}^{(p)}))$
7:    // *Weight transfer*
8:    $\hat{\boldsymbol{\mathcal{G}}}^{(k)} \leftarrow \boldsymbol{\mathcal{G}}^{(k)}$ for $k \in [p] \setminus \{i, j\}$;    $R_{i,j} \leftarrow R_{i,j} + 1$
9:    $\hat{\boldsymbol{\mathcal{G}}}^{(i)} \leftarrow$ `add-slice`$(\boldsymbol{\mathcal{G}}^{(i)}, j)$   // *add new slice to the jth mode of $\boldsymbol{\mathcal{G}}^{(i)}$*
10:   $\hat{\boldsymbol{\mathcal{G}}}^{(j)} \leftarrow$ `add-slice`$(\boldsymbol{\mathcal{G}}^{(j)}, i)$   // *add new slice to the ith mode of $\boldsymbol{\mathcal{G}}^{(j)}$*
11:   // *Optimize new tensor network structure*
12:   $(\boldsymbol{\mathcal{G}}^{(1)}, \cdots, \boldsymbol{\mathcal{G}}^{(p)}) \leftarrow$ `optimize` $\mathcal{L}(\text{TN}(\boldsymbol{\mathcal{G}}^{(1)}, \cdots, \boldsymbol{\mathcal{G}}^{(p)}))$ from init. $\hat{\boldsymbol{\mathcal{G}}}^{(1)}, \cdots, \hat{\boldsymbol{\mathcal{G}}}^{(p)}$
13:   // *Add internal nodes if possible (number of cores p may be increased after this step)*
14:   $(\boldsymbol{\mathcal{G}}^{(1)}, \cdots, \boldsymbol{\mathcal{G}}^{(p)}) \leftarrow$ `split-nodes`$((\boldsymbol{\mathcal{G}}^{(1)}, \cdots, \boldsymbol{\mathcal{G}}^{(p)}), \varepsilon)$
15: **until** Stopping criterion

---

be $d_1 \times (R_{1,2} + 1) \times R_{1,3} \times R_{1,4}$ and the one of $\boldsymbol{\mathcal{G}}^{(2)}$ will be $(R_{1,2} + 1) \times d_2 \times R_{2,3} \times R_{2,4}$. The following proposition shows that if these slices were initialized exactly to 0, the resulting TN would represent exactly the same tensor as the original one. In practice, we initialize the slices randomly with small values to break symmetries that could constrain the continuous optimization process.

**Proposition 1.** *Let $\boldsymbol{\mathcal{G}}^{(k)} \in \mathbb{R}^{R_{1,k} \times \cdots \times R_{k-1,k} \times d_k \times R_{k,k+1} \times \cdots \times R_{k,p}}$ for $k \in [p]$ be the core tensors of a tensor network and let $1 \le i < j \le p$. Let $\tilde{R}_{i'j'} = R_{i',j'} + 1$ if $(i', j') = (i, j)$ and $R_{i',j'}$ otherwise, and define the core tensors $\tilde{\boldsymbol{\mathcal{G}}}^{(k)} \in \mathbb{R}^{\tilde{R}_{1,k} \times \cdots \times \tilde{R}_{k-1,k} \times d_k \times \tilde{R}_{k,k+1} \times \cdots \times \tilde{R}_{k,p}}$ for $k \in [p]$ by*

$$(\tilde{\boldsymbol{\mathcal{G}}}^{(i)})_{(j)} = \begin{bmatrix} (\boldsymbol{\mathcal{G}}^{(i)})_{(j)} \\ -\mathbf{0}- \end{bmatrix}, \ (\tilde{\boldsymbol{\mathcal{G}}}^{(j)})_{(i)} = \begin{bmatrix} (\boldsymbol{\mathcal{G}}^{(j)})_{(i)} \\ -\mathbf{0}- \end{bmatrix} \ and \ \tilde{\boldsymbol{\mathcal{G}}}^{(k)} = \boldsymbol{\mathcal{G}}^{(k)} \ for \ k \in [p] \setminus \{i, j\}$$

*where $\mathbf{0}$ denotes a row vector of zeros of the appropriate size in each block matrix.*

*Then, the core tensors $\tilde{\boldsymbol{\mathcal{G}}}^{(k)}$ correspond to the same tensor network as the core tensors $\boldsymbol{\mathcal{G}}^{(k)}$, i.e., $TN(\tilde{\boldsymbol{\mathcal{G}}}^{(1)}, \cdots, \tilde{\boldsymbol{\mathcal{G}}}^{(p)}) = TN(\boldsymbol{\mathcal{G}}^{(1)}, \cdots, \boldsymbol{\mathcal{G}}^{(p)})$.*

The proof of the proposition can be found in Appendix A. The weight transfer mechanism leads to a more efficient and robust continuous optimization by transferring the knowledge from each greedy iteration to the next and avoiding re-optimizing the loss function from a random initialization at each iteration. An ablation study showing the benefits of weight transfer is provided in Appendix C.3.

**Best Edge Selection** As mentioned previously, we propose to optimize the inner minimization problem in Eq. 2 using iterative algorithms, namely gradient based algorithms or ALS depending on the loss function $\mathcal{L}$. In order to identify the most promising edge to increase the rank by 1 (line 6 of Algo. 1), a reasonable heuristic consists in optimizing the loss for a few epochs/iterations for each possible edge and select the edge which led to the steepest decrease in the loss. One drawback of this approach is its computational complexity: e.g., for ALS, each iteration requires solving $p$ least-squares problem with $d_i \prod_{k \neq i} R_{i,k}$ unknowns for $i \in [p]$. We propose to reduce the complexity of the exploratory optimization in the best edge identification heuristic by only optimizing $\mathcal{L}$ w.r.t. the new slices of the core tensors. Thus, at each iteration of the greedy algorithm, for each possible edge to increase, we transfer the weights from the previous greedy iteration, optimize only w.r.t. the new slices for a small number of iteration, and choose the edge which led to the steepest decrease of the loss. For ALS, this reduces the complexity of each iteration to the one of solving 2 least-squares problems with $d_i \prod_{k \in [p] \setminus \{i,j\}} R_{i,k}$ and $d_j \prod_{k \in [p] \setminus \{i,j\}} R_{i,k}$ unknowns, respectively, where $(i, j)$ is the edge being considered in the search. When using gradient-based optimization algorithms, the same approach is used where the gradient is only computed for (and back-propagated through) the

new slices. It is worth mentioning that the greedy algorithm can seamlessly incorporate structural constraints by restricting the set of edges considered when identifying the best edge for a rank increment. For example, it can be used to adaptively select the ranks of a TT or TR decomposition.

**Internal Nodes**   Lastly, we design a simple approach for the greedy algorithm to add internal nodes to the TN structure relying on a common technique used in TN methods to split a node into two new nodes using truncated SVD (see, e.g., Fig. 7.b in [35]). To illustrate this technique, let $\mathcal{M} \in \mathbb{R}^{m_1 \times m_2 \times n_1 \times n_2}$ be the core tensor associated with a node in a TN we want to split into two new nodes $\mathcal{A} \in \mathbb{R}^{m_1 \times m_2 \times r}$ and $\mathcal{B} \in \mathbb{R}^{n_1 \times n_2 \times r}$: the first two legs of $\mathcal{A}$ (resp. $\mathcal{B}$) will be connected to the core tensors that were connected to $\mathcal{M}$ by its first two legs (resp. last two legs), and the third leg of $\mathcal{A}$ and $\mathcal{B}$ will be connected together. This is achieved by taking the rank $r$ truncated SVD of $(\mathcal{M})_{(1,2)} \simeq \mathbf{U} \mathbf{D} \mathbf{V}^\top \in \mathbb{R}^{m_1 m_2 \times n_1 n_2}$ (the matricization of $\mathcal{M}$ having modes 1 and 2 as rows and modes 3 and 4 as columns), and letting $\mathcal{A}_{(3)} = \mathbf{U}^\top \in \mathbb{R}^{r \times m_1 m_2}$ and $\mathcal{B}_{(3)} = \mathbf{D} \mathbf{V}^\top \in \mathbb{R}^{r \times n_1 n_2}$. If the truncated SVD is exact, the resulting TN will represent exactly the same tensor as the one before splitting the core $\mathcal{M}$. This node splitting procedure is illustrated in the following TN diagram.

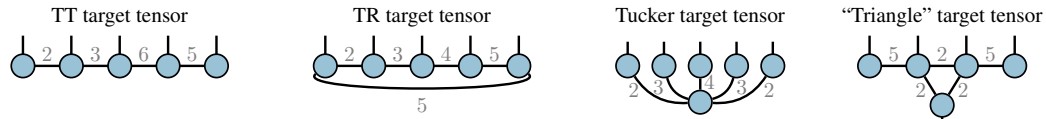

In order to allow the greedy algorithm to learn TN structures with internal nodes, at the end of each greedy iteration, we perform an SVD of each matricization of $\mathcal{G}^{(k)}$ for $k \in [p]$ (line 14 of Algo. 1). For each matricization, we split the corresponding node only if there are enough singular values below a given threshold $\varepsilon$ in order for the new TN structure to have less parameters than the initial one. While this approach may seem computationally heavy, the cost of these SVDs is negligible w.r.t. the continuous optimization step which dominates the overall complexity of the greedy algorithm.

The details of all subroutines of Algorithm 1 as well as its complexity are given in the Appendix B.

## 4   EXPERIMENTS

In this section we evaluate `Greedy-TN` on decomposition, completion, and model compression.

**Tensor decomposition**   We first consider a tensor decomposition task. We randomly generate four target tensors of size $7 \times 7 \times 7 \times 7 \times 7$ with the following TN structures:

| TT target tensor | TR target tensor | Tucker target tensor | "Triangle" target tensor |

We run `Greedy-TN` until it recovers an almost exact decomposition (stopping when the relative error falls below $10^{-6}$). We compare `Greedy-TN` with CP, Tucker and TT decomposition (using the implementations from the TensorLy python package [18]) of increasing rank as baselines (we use uniform ranks for Tucker and TT). We also include a simple random walk baseline based on `Greedy-TN`, where the edge for the rank increment is chosen at random at each iteration. Reconstruction errors averaged over 100 runs of this experiment are reported in Figure 4, where we see that the greedy algorithm outperforms all baselines for the the four target tensors. Notably, `Greedy-TN` outperforms TT/Tucker even on the TT/Tucker targets. This is because the rank of the TT and Tucker targets are not uniform and `Greedy-TN` is able to adaptively set different ranks to achieve the best compression ratio. Furthermore, `Greedy-TN` is able to recover the exact TN structure of the triangle target tensor on almost every run. Lastly, we observe that the internal node search of `Greedy-TN` is only beneficial on the Tucker target tensor, which is expected due to the absence of internal nodes in the other target TN structures. As an illustration of the running time, for the TR target, one iteration of `Greedy-TN` takes approximately 0.91 second on average without the internal node search and 1.18 seconds with the search. The most common TN structures recovered by `Greedy-TN` are shown in Appendix C.1. This experiment showcases the potential cost of model mis-specification: both CP and Tucker struggle to efficiently approximate most target tensors. Interestingly, even the random walk outperforms CP and Tucker on the TR target tensor.

**Tensor completion**   We compare `Greedy-TN` with the TT and TR alternating least square algorithms proposed in [37] and the CP and Tucker decomposition algorithms from Tensorly [18] on an

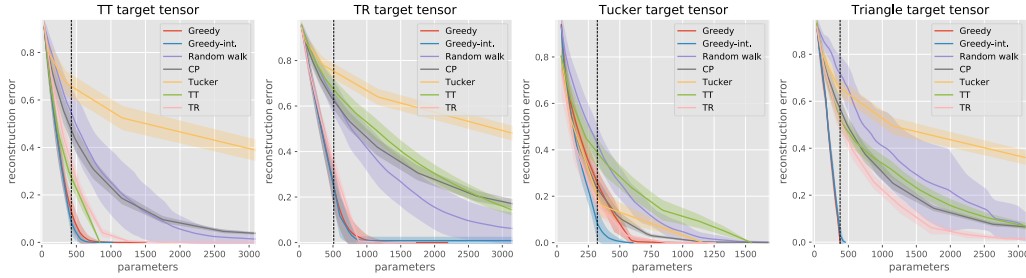

Figure 4: Evaluation of `Greedy-TN` on tensor decomposition. Curves represent the reconstruction error averaged over 100 runs, shaded areas correspond to standard deviations and the vertical line represents the number of parameters of the target TN. `Greedy` corresponds to `Greedy-TN` without the search for internal nodes (line 14 of Algo. 1) while `Greedy-int.` includes this search.

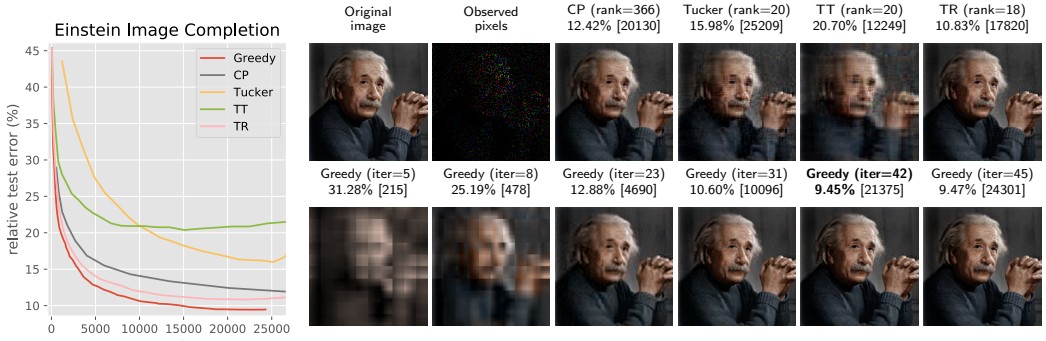

Figure 5: Image completion with 10% of the entries randomly observed. (left) Relative reconstruction error. (right) Best recovered images for CP, Tucker, TT and TR, and 6 recovered images at different iteration of greedy (image title: RSE% [number of parameters]).

image completion task. We consider an experiment presented in [37]: the completion of an RGB image of Albert Einstein reshaped into a $6 \times 10 \times 10 \times 6 \times 10 \times 10 \times 3$ tensor (see [37] for details) where 10% of entries are randomly observed. The ranks of methods other than `Greedy-TN` are successively increased by one until the number of parameters gets larger than 25,000 (we use uniform ranks for TT, TR and Tucker[*]). The relative errors as a function of number of parameters are reported in Figure 5 (left) where we see that `Greedy-TN` outperforms all methods. The best recovered images for all methods are shown in Figure 5 (right) along with the original image and observed pixels. The best recovery error (9.45%) is achieved by `Greedy-TN` at iteration 42 with 21,375 parameters. The second best recovery error (10.83%) is obtained by `TR-ALS` at rank 18 with 17,820 parameters. At iteration 31, `Greedy-TN` already recovers an image with an error of $10.60\%$ with 10,096 parameters, which is better than the best result of `TR-ALS` both in terms of parameters and relative error. The images recovered at each iteration of `Greedy-TN` are shown in Appendix C.2. In this experiment, the total running time of `Greedy-TN` is comparable to the one of `TR-ALS` (on the order of hours), which is larger than the one of the other three methods.

**Image compression** In this experiment, we compare `Greedy-TN` with the genetic algorithm for TN decomposition recently introduced in [19], denoted by GA(rank=6) and GA(rank=7) where the rank is a hyper-parameter controlling the trade-off between accuracy and compression ratio (the results of TT and TR, which are worst than GA, are available in Table 3 in [19]). Following [19], we select 10 images of size $256 \times 256$ from the LIVE dataset [32], tensorize each image to an order-8 tensor of size $4^8$ and run `Greedy-TN` to decompose each tensor using a squared error loss. `Greedy-TN` is stopped when the lowest RSE reported in [19] is reached. In Table 1, we report the log compression ratio and root square error averaged over 50 random seeds. For all images,

---

[*]For Tucker, the completion is performed on the original image rather than the tensor reshaping since the number of parameters of Tucker grows exponentially, leading to very poor results on the tensorized image.

| Image | Log compression ratio CR↑ and (RSE↓) ±std | | |
|---|---|---|---|
| | Greedy | GA(rank=6) | GA(rank=7) |
| 0 | **0.715(0.105)**±0.152(0.005) | **0.901(0.137)** | 0.660(0.115) |
| 1 | **2.313(0.150)**±0.189(0.005) | 1.352(0.158) | 1.159(0.155) |
| 2 | **2.139(0.167)**±0.127(0.004) | 1.452(0.176) | 1.268(0.171) |
| 3 | **3.009(0.185)**±0.088(0.002) | 1.649(0.193) | 1.476(0.189) |
| 4 | **0.874(0.111)**±0.129(0.005) | 0.859(0.152) | 0.621(0.121) |
| 5 | **3.668(0.080)**±0.103(0.001) | 1.726(0.087) | 1.548(0.083) |
| 6 | **2.205(0.097)**±0.171(0.004) | 1.332(0.110) | 1.141(0.104) |
| 7 | **2.132(0.115)**±0.202(0.002) | 1.573(0.126) | 1.406(0.120) |
| 8 | **3.634(0.080)**±0.142(0.001) | 1.679(0.085) | 1.505(0.081) |
| 9 | **1.669(0.174)**±0.202(0.002) | 1.164(0.194) | 0.966(0.185) |

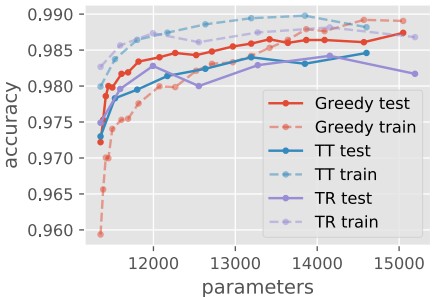

Table 1: Log compression ratio and RSE for 10 different images selected from the LIVE dataset.

Figure 6: Train and test accuracies on the MNIST dataset for different model sizes.

our method results in a higher compression ratio compared to GA(rank=7). Moreover, for images 1 to 9 our method even outperforms GA(rank=6) by achieving both higher compression ratios and significantly lower RSE. For image 0, setting the greedy stopping criterion to the RSE of GA(rank=6), `Greedy-TN` also achieves a higher compression ratio than GA(rank=6): $1.085(0.128)$. Our method is also orders of magnitude faster—few minutes compared to several hours for GA.

**Compressing neural networks** Following [24], we apply our algorithm to compress a neural network with one hidden layer on the MNIST dataset. The hidden layer is of size $1024 \times 1024$ which we represent as a fifth-order tensor of size $16 \times \cdots \times 16$. We use `Greedy-TN` to train the core tensors representing the hidden layer weight matrix alongside the output layer end-to-end. We select the best edge for the rank increment using the validation performance on a separate random split of the train dataset with $5,000$ images. In Figure 6, we report the train and test accuracies of the TT based method introduced in [24] as well as a TR tensorized model for uniform ranks 1 to 8 and `Greedy-TN` (it is worth noting that we use our own implementation of the TT method with dropout and achieve higher accuracies than the ones reported in [24]). For every model size, our method reaches higher accuracy. The best test accuracy of `Greedy-TN` is $98.74\%$ with 15,050 parameters, while TT reaches its best accuracy of $98.46\%$ with 14,602 parameters, and TR achieves its best accuracy of $98.42\%$ with 14,154 parameters. At iteration 10, `Greedy-TN` already achieves an accuracy of $98.46\%$ with only 12,266 parameters. The running time of each iteration of `Greedy-TN` is comparable with training one tensorized neural network with TT or TR.

**Implementation details** We use PyTorch [27] and the NCON function [30] to implement `Greedy-TN`. For the continuous optimization step, we use the Adam [16] optimizer with a learning rate of $10^{-3}$ and a batch size of $256$ for $50$ epochs for compressing neural network, and we use ALS for the other three experiments (ALS is stopped when convergence is reached). The number of iterations/epochs for the best edge identification is set to 2 for tensor decomposition, 5 for image compression and 10 for image completion and compressing neural networks. The singular values threshold for the internal node search is set to $\varepsilon = 10^{-5}$. In all experiments except the tensor decomposition on the Tucker target, the internal node search did not lead to any improvement of the results. All experiments were performed on a single 32GB V100 GPU.

## 5 CONCLUSION

We introduced a greedy algorithm to jointly optimize an arbitrary loss function and efficiently search the space of TN structures and ranks to adaptively find parameter efficient TN structures from data. Our experimental results show that `Greedy-TN` outperforms common methods tailored for specific decomposition models on model compression and tensor decomposition and completion tasks. Even though `Greedy-TN` is orders of magnitude faster than the genetic algorithm introduced in [19], its computational complexity can still be limiting in some scenarios. In addition, the greedy algorithm may converge to locally optimal TN structures. Future work includes exploring more efficient discrete optimization techniques to solve the upper-level discrete optimization problem and scaling up the method to discover TN structures suited for efficient compression of larger neural network models.

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
