# OpenReview forum: "Adaptive Learning of Tensor Network Structures"
_ICLR.cc/2022/Conference — ICLR 2022 Submitted_

### Official Review · Reviewer_uoVk · 2021-10-20

**Correctness:** 2
**Technical Novelty And Significance:** 2
**Empirical Novelty And Significance:** 2
**Recommendation:** 5
**Confidence:** 3

**Main Review:**

The main idea is a bit ad-hoc: gradually increasing the dimension of the core tensor, where the factor matrices & core tensor must be recalculated/optimized as soon as the dimension of the core tensor changes. It is obvious that such an optimization method requires very high complexity, and it's hard to have convergence guarantee.

The authors claimed that the TN also works for CP decomposition. However, the procedures in Algorithm 1 actually may not support their claim. Because there is no such a so-called core tensor in CPD. What I can image is that the core tensor in CPD is a N-D diagonal tensor. In such a case, how does Algorithm 1 gradually increase one slab to the core tensor (which will destroy the N-D diagonal structure), and guarantee the core tensor is diagonal?

The description of Algorithm is a bit unclear. In the Best Edge Selection, it says the gradient or LS is only applied to the new slice, then how to guarantee the objective to decrease without taking the remaining slices into consideration? Obviously, this is very suboptimal.

Lastly, to be honest, I didn't understand the section of Internal Nodes.

**Summary Of The Paper:**

This paper proposed an adaptive method that enables to optimize the best tensor network structure which can be used for tensor completion, decomposition, etc. The performance of the proposed method was tested on both synthetic and real data.

**Summary Of The Review:**

Based on my own understanding, the method is not well explained. It is ad-hoc and no theoretical guarantee of its convergence. I would not recommend it for publication.

---

### Official Review · Reviewer_RmTg · 2021-11-01

**Correctness:** 3
**Technical Novelty And Significance:** 2
**Empirical Novelty And Significance:** 2
**Recommendation:** 3
**Confidence:** 4

**Main Review:**

Strengths

1. This paper focuses on the problem of the structure determination of tensor networks, which is an important topic in tensor learning literature.
2. The paper is clearly written and easy to follow.
3. A practical algorithm is proposed.


Weaknesses

1. The practical values of the proposed method are not clearly shown. In the experiments, the method is applied to image and neural network (NN) compression tasks, but there are strong competitors for their applications (discrete cosine transform for image and distillation/neural architecture search for NN). Do you have any other specific applications where the proposed method will win? If not, it would be better to other venues such as linear algebra rather than machine learning.
2. The proposed algorithm is based on heuristics and there is no theoretical guarantee. Greedy algorithms can have a property that allows evaluating the error between the optimal and obtained solutions for specific objectives (e.g. submodular function) [*]. Could you provide such an analysis?
3. The paper is not self-contained. In Algorithm 1, the definition of the subfunction `split-nodes` is not described. The stopping criterion is also not described.
4. It is not clearly discussed what kind of class of tensor networks is considered. In the 2nd paragraph of Section 3.1, it is said `Without loss of generality, we consider TN having one factor per dimension ...`. But this class doesn't include tensor networks in which a factor has two or more output legs or tensor networks that have hyper-edges (i.e. contraction more than two indices). I believe this violates "the loss of generality".


[*] https://home.csulb.edu/~tebert/teaching/lectures/528/greedy/greedy.pdf

**Summary Of The Paper:**

In this paper, a greedy algorithm that can find a structure of a certain class of tensor networks is proposed. The algorithm consists of bi-level optimization, where tensor network structure is optimized in the outer loop and tensor decomposition is computed to approximate a given tensor in the inner loop. The tradeoff between error and the number of parameters of the proposed algorithm is empirically compared with synthetic data, image compression, and neural network compression.

**Summary Of The Review:**

This paper tackles an important problem and provides a handy algorithm. However, the impact of this study is not significantly presented in the machine learning context. The technical contributions are based on heuristics and not technically solid.

---

### Official Review · Reviewer_yrur · 2021-11-02

**Correctness:** 2
**Technical Novelty And Significance:** 2
**Empirical Novelty And Significance:** 1
**Recommendation:** 3
**Confidence:** 5

**Main Review:**

The greedy method to find the structure or rank of TN and algorithms included ALS, DMRG ...  for TNs were presented there.
Many similar algorithms or variants have been recently proposed,  e.g., "Adaptive Rank Selection for Tensor Ring Decomposition. 2021.

In addition, the tensor ring is indeed the tensor chain model which was proposed in 2009, 2011 by Khoromskij. The authors should give honor to the authors who first invented the model. For computations in Tensor chain and tensor networks, see Espig, Hackbusch, Handschuh and Schneider (2012a), Espig, Naraparaju, and  Schneider 2012, Huckle, Waldherr and Schulte-Herbruggen (2013), Hackbusch 2014, and Handschuh, 2015. See also Cichocki et al 2016, 2018.
Some other publications later follow the new name of the TC model and may not be aware of the original work of the Tensor chain and algorithms for generalized TNs.

The authors use the ALS algorithm as the core algorithm for TN decomposition. This algorithm, especially for the CP and TC models, often gets stuck in false local minima and prevents the greedy method from finding a good model.

The authors show that they can recover the exact TN structure of the synthetic tensor of size 7 x 7 x 7 x 7 x 7, which admits the TC model of bond dimensions 2-3-6-5. For this example, the core tensors were generated from a normal distribution, and the TC tensor is relatively easy to decompose.
The authors can try the TC decomposition for the tensor of the same tensor size and bond dimensions (3-3-3-3), but the cores are generated from a uniform distribution. The latter tensor is even more challenging for TC decomposition, and most algorithms succeeded with a rate lower than 10\%.

This is to show that most greedy algorithms cannot find the "true" model of a tensor (the problem is indeed NP-hard), even with known ranks.

I also suggest the authors test their method for tensors associated with the multiplication of two matrices.


For TT, the TT-SVD or better algorithms for TT with bounded approximation error should be used.
The performance for TT shown in Figure 5 seems incorrect. TT with higher number parameters should give better reconstruction or lower relative error.

An example of compression of neural networks is for a very simple network with only one hidden layer and the MNIST dataset. It is known that the MNIST dataset is easy to fit and obtain excellent accuracy. The authors should try, e.g., Resnet for Imagnet dataset or CIFAR 10, CIFAR100.


- References [17, 5] do not present ALS algorithm for Tensor network,
but Espig, Hackbusch, Handschuh, and  Schneider 2012.

- The implementation provided in the supplementary does not support the TC model.


**Summary Of The Paper:**

First, the idea of rank incremental method for Tensor network decomposition and determination of TN structures is not novel.
Second, decomposition with weight transfer is obvious and widely used in the rank incremental method.
For generalized tensor networks decomposition, the authors should consider the works.

S.  Handschuh,  “Numerical  Methods  in  Tensor  Networks,” PhD  thesis,Facualty of Mathematics and Informatics, University Leipzig, Germany,Leipzig, Germany, 2015.

Mike Espig, Wolfgang Hackbusch, Stefan Handschuh, and Reinhold Schneider,   Optimization Problems in Contracted Tensor Networks, 2012

**Summary Of The Review:**

The proposed method is not novel, in two contributions that authors claimed: greedy method and sharing weights

Decompositions such as with CPD and TC very often encounter degeneracy and cannot find the true model even with true ranks.

NNs and the considered dataset for the compression task is relatively simple and easy to compress.
This does not confirm the greedy algorithm works well for other difficult scenarios.

---

### Official Review · Reviewer_7PSK · 2021-11-05

**Correctness:** 3
**Technical Novelty And Significance:** 3
**Empirical Novelty And Significance:** 3
**Recommendation:** 5
**Confidence:** 4

**Details Of Ethics Concerns:**

No Ethics Concerns.

**Main Review:**

This paper is interesting in that it presents a greedy algorithm for learning tensor network structure. The paper also discusses the relationship among different tensor representation, and the authors have a deep understanding of tensor decomposition an optimization problems. My concerns mostly lie in the design of the algorithm and experiments.

1. How is the core tensor randomly initialized in Algorithm 1?

2. It may be inappropriate to claim that the Tucker decomposition is not well-suited for tensors of very high order. Would reshaping a third-order tensor (RGB image) into a very high-order tensor leads to certain loss of structure information?

3. The result of the image completion is not very convincing since the authors only perform experiments on one sample image with a fixed missing ratio. More experimental results should be conducted and some recently proposed tensor-based methods should also be considered. For example, "Image Completion Using Low Tensor Tree Rank and Total Variation Minimization", "Tensor Completion via Nonlocal Low-Rank Regularization".

4. Could the authors also compare the computational complexity or list detailed running/training time of compared methods?

5. Is the proposed method subject to the influence of noise (AWGN)? Can it be applied to the image denoising task where Tucker decomposition shows very good performance?

**Summary Of The Paper:**

This paper proposes a greedy algorithm to solve the tensor network optimization problem in a heuristic manner. The major contribution of the method is the greedy algorithm to efficiently derive the tensor network structure.

**Summary Of The Review:**

The paper is technically interesting, but the authors should carefully revise their experimental section to address the above concerns.

---

### Decision · Program_Chairs · 2022-01-20

**Decision:**

Reject

**Comment:**

The paper considers the important problem of tensor network optimization. Unfortunately the authors did not respond to the reviewers comments. Hence, several concerns remain about the proposed greedy algorithm, including its relationship with prior work and the issue of the ALS method being stuck in local minima for important classes of problems. We strongly encourage the authors to carefully examine the reviewers points and revise their work accordingly.